# Effects of Equol Supplementation on Growth Performance, Redox Status, Intestinal Health and Skeletal Muscle Development of Weanling Piglets with Intrauterine Growth Retardation

**DOI:** 10.3390/ani13091469

**Published:** 2023-04-26

**Authors:** Yong Zhang, Jingchang Ren, Li Chen, Honglin Yan, Tiande Zou, Hongfu Zhang, Jingbo Liu

**Affiliations:** 1School of Science and Engineering, Southwest University of Science and Technology, Mianyang 621010, China; zyzlrzjh@swust.edu.cn (Y.Z.); 15983100307@163.com (J.R.); 15681159587@163.com (L.C.); honglinyan@swust.edu.cn (H.Y.); 2Jiangxi Province Key Laboratory of Animal Nutrition, Jiangxi Agricultural University, Nanchang 330045, China; tiandezou@jxau.edu.cn; 3Institute of Animal Sciences, Chinese Academy of Agricultural Sciences, Beijing 100193, China; zhanghfcaas@gmail.com

**Keywords:** equol, intrauterine growth retardation, intestinal health, skeletal muscle development, piglets, redox status

## Abstract

**Simple Summary:**

Intrauterine growth retardation (IUGR) hurts the postnatal growth and development of weanling piglets. Equol (Eq), a primary bioactive metabolite of daidzein, derives from intestinal bacterial metabolism and exerts numerous biological benefits. Nevertheless, no evidence is available to discover whether dietary Eq treatment exerts positive influences on the growth performance, redox status, intestinal health and skeletal muscle development of piglets with IUGR. Therefore, the aim of this study was to evaluate the influences of Eq supplementation on growth performance, redox status, intestinal health and skeletal muscle development. Here, twenty IUGR piglets and ten normal-birth-weight (NBW) female weanling piglets were used in this study and the results showed that Eq treatment enhanced antioxidant capacity and intestinal health, and facilitated skeletal muscle development, thus promoting the growth performance of IUGR piglets. Our findings provide significant implications for improving the growth performance of IUGR piglets and highlighting feasible applications in pig production.

**Abstract:**

Animals with intrauterine growth retardation (IUGR) usually undergo injured postnatal growth and development during the early period after birth. Equol (Eq), an isoflavan produced by gut bacteria in response to daidzein intake, has various health benefits. Therefore, the objective of this study was to evaluate whether Eq supplementation can influence the growth performance, redox status, intestinal health and skeletal muscle development of weanling piglets with IUGR. A total of 10 normal-birth-weight (NBW) newborn female piglets and 20 newborn female piglets with IUGR were selected. After weaning at the age of 21 d, 10 NBW piglets and 10 IUGR piglets were allocated to the NBW group and IUGR group, respectively, and offered a basal diet. The other 10 IUGR piglets were allocated to the IUGR + Eq group and offered a basal diet with 50 mg of Eq per kg of diet. The whole trial lasted for 21 d. At the end of the feeding trial, all piglets were sacrificed for the collection of serum, intestinal tissues and skeletal muscles. Supplementation with Eq increased the average daily gain (ADG), average daily feed intake (ADFI), duodenal villus height to crypt depth ratio (V/C), jejunal villus height and V/C, but reduced the duodenal crypt depth in neonatal piglets with IUGR. Meanwhile, Eq supplementation elevated the activities of superoxide dismutase (SOD) and catalase (CAT) in the serum and duodenum and the activity of SOD in the jejunum, but lowered malondialdehyde (MDA) content in the serum, jejunum and ileum of piglets with IUGR. In addition, supplementation with Eq reduced diamine oxidase (DAO) activity and the levels of D-lactate and endotoxin in serum, and the tumor necrosis factor-α (TNF-α) level in jejunum and ileum, whereas the concentration of serum immunoglobulin G (IgG) and the mRNA levels of intestinal barrier-related markers in jejunum and ileum of IUGR piglets were increased. Furthermore, supplementation with Eq elevated the percentage of fast-fibers and was accompanied with higher mRNA expression of myosin heavy chain IIb (*MyHC IIb*) and lower mRNA levels in *MyHC I* in the longissimus thoracis (LT) muscle of IUGR piglets. In summary, Eq supplementation can promote antioxidant capacity, maintain intestinal health and facilitate skeletal muscle development, thus resulting in the higher growth performance of IUGR piglets.

## 1. Introduction

Intrauterine growth retardation (IUGR) generally refers to the abnormal growth and development of the embryo/fetus or its organs during the gestation period [1]. Neonates with IUGR exhibit oxidative damage, gut dysfunction and reduced muscular growth, which has a perpetual stunting influence on postnatal growth, nutrient availability and the normal function of organs or tissues [1,2,3], and weanling may further aggravate negative influences on the growth and development of IUGR. As a domestic multiparous animal, pigs often suffer a high incidence of IUGR (approximately 15–20%), which would lead to considerable economic losses in the swine industry [4]. This condition is more common in highly prolific sows because of the competition among embryos for the uterine space, which may lead to an insufficiency of placental growth and inadequate provision of nutrients and oxygen to the fetuses, thus causing more IUGR newborns [5]. Therefore, it is urgently necessary to develop new strategies to improve the postnatal growth and development of IUGR piglets.

Specific nutritional administration provides a feasible method to relieve the complications of IUGR piglets [3,6]. Daidzein, one of the most common isoflavones in leguminous plants, is well-known to show a broad series of biological properties, such as immunomodulatory, antioxidant, anticancer, antimicrobial and estrogenic activities [7,8,9,10]. Interestingly, several research works have implied that the health benefits of isoflavones might be reliant on their metabolites [11,12,13]. Equol (Eq), a heterocyclic phenol, is a metabolite daidzein-derived by bacteria in the distal ileum and colon [14]. Eq possesses strong estrogenic, antiandrogenic, antioxidant and anti-inflammatory properties [15,16,17], as well as cell cycle regulatory activity [18]. Among these biological activities, the antioxidative capacity of Eq has recently gained considerable attention in animal husbandry [19] and was found to be higher than that of daidzein [20]. In vitro, Eq pretreatment protects primary chicken intestinal epithelial cells from oxidative damage induced by H_2_O_2_ via elevating the expressions of antioxidant-related genes and enzymes [21]. In vivo, equol treatment markedly suppressed oxidative stress-related markers and promoted the activities of catalase (CAT) and total superoxide dismutase (T-SOD) and their mRNA expression in the livers of mice [17]. Moreover, Eq can block lipopolysaccharide-induced oxidative stress and promote immunoreaction in chicken HD11 macrophages [16]. In addition, Eq is more easily absorbed, more stable, and has a lower clearance than its precursor, daidzein [22].

However, no animal experiment has evaluated the influence of Eq treatment on the growth performance and aspects of intestinal health and skeletal muscle development of IUGR piglets. Considering its potent antioxidant ability and other biological functions, it was supposed that Eq may be an effective bioactive compound to relieve the maldevelopment related to IUGR. Hence, the aim of the current study was to investigate the influences of Eq on growth performance, redox status, intestinal health and muscle development of weanling piglets with IUGR.

## 2. Materials and Methods

### 2.1. Animals and Treatments

Ten healthy, pregnant sows with the third to fifth parity were selected until delivery. During gestation, a vaccination program against *Porcine reproductive and respiratory syndrome virus* (PPRSV), *Swine influenza virus*, *Porcine parvovirus*, *Escherichia coli* and *Swine streptococcosis* was implemented. After delivery at day 114, the reproductive performance of sows was as follows: the number of total born piglets was 11.89 ± 1.05, 11.02 ± 1.12 were live births, and the number of stillborn and mummified piglets was 1.07 ± 0.85. One NBW female piglet [Duroc × (Landrace × Yorkshire)] with mean body weight (BW) of 1.53 ± 0.09 kg and two IUGR female piglets with mean BW of 0.89 ± 0.06 kg were chose from same litter according to the criteria of D’Inca et al. [23]. All piglets were reared following the standard feeding procedure until weaning at 21 ± 1 d of age [24,25]. Then, NBW piglets were allocated to the NBW group, while two IUGR piglets from same litter with similar body weight were randomly allocated to the IUGR group or IUGR + Eq group (n = 10). Piglets in NBW and IUGR groups were offered a basal diet, and piglets in the IUGR + Eq group were offered a basal diet supplemented with 50 mg of Eq per kg of diet (purity ≥ 99%, Zhengzhou Acme Chemical Co., Ltd., Henan, China). The addition level of Eq in this trial was tested following the results of a preliminary trial, where a higher growth rate and lower diarrhea incidence were observed in weaned piglets when Eq was supplied at 50 mg of Eq per kg of diet (unpublished).

### 2.2. Diets and Feeding Management

The basal diet was created following the National Research Council (NRC, 2012) recommendation for weaned piglets, and did not contain antibiotics or other growth promoters. The formulation and calculated composition of the basal diet are exhibited in Table 1. All piglets were fed separately in a temperature-controlled room. The feed and water were free to access. No treatment was adopted if diarrhea occurred. After adapting for 3 d, piglets were supplied their respective diets four times in one day. The whole experiment lasted for 28 d. The temperature of the enclosure was sustained at 30 °C and the relative humidity was maintained at 50–60%. On d 1 and d 28, all piglets were weighed individually after 12 h fasting, and the feed consumption of every cage was marked daily during the whole trial to evaluate the growth performance indicators.

### 2.3. Sample Collection

At the end of the feeding trial, blood was harvested into a 10 mL sterile vacutainer tube without any anticoagulant by jugular venipuncture after overnight fasting. After centrifugation at 4 °C for 10 min (3500× *g*), supernatants were collected and conserved at –20 °C until determination. Subsequently, all piglets were slaughtered with a lethal dose of sodium pentobarbital (200 mg/kg BW) by intravenous injection [27]. Subsequently, small intestinal samples and longissimus thoracis (LT) muscles were collected for further analysis.

### 2.4. Intestinal Morphology Analysis

Firstly, about 2 cm portions of different small intestines were collected and fixed in 10% formalin solution and dehydrated using increasing grades of ethanol. Subsequently, the dried intestines underwent the process of embedding, sectioning and staining. The indicators for intestinal morphology included villus height, crypt depth, villus width and villus surface area (VSA). The villus height was measured from the tip to the base, the crypt depth was detected from the crypt–villus junction to the base, and the villus area was the distance of the widest villus. The VSA could be calculated according to Bai et al. [28] and as follows:$$=\pi \times \left(\frac{\mathrm{villus}\;\mathrm{width}}{2}\right)\;\sqrt{{\left(\frac{\mathrm{villus}\;\mathrm{width}}{2}\right)}^{2}+{\mathrm{villus}\;\mathrm{height}}^{2}}$$

### 2.5. Analysis of Antioxidant-Related Indices

The frozen mucosa samples from different intestinal segments were weighed (about 1.0 g), ground with a hand-held tissue homogenizer (Fisher, Pittsburgh, PA, USA) in 9.0 mL cold physiological saline and then centrifuged at 3500 r/min for 15 min at 4 °C to harvest the supernatants. Malondialdehyde (MDA) content, and the activities of glutathione peroxidase (GSH-Px), superoxide dismutase (SOD), catalase (CAT), and total antioxidant capacity (T-AOC) were tested using a microplate reader (BioRad, Hercules, CA, USA) with their kits (Nanjing Jiancheng Bioengineering Institute, Nanjing, China) following kit protocols. The concentration of total protein in mucosa homogenates was tested using the bicinchoninic acid (BCA) method for normalization.

### 2.6. Serum Parameter Measurement

Serum D-lactate concentration and diamine oxidase (DAO) activity were detected using kits (Sino-German Beijing Leadman Biotech Ltd., Beijing, China and Nanjing Jiancheng Bioengineering Institute, Nanjing, China) following the manufacturer protocols. The serum endotoxin level was tested spectrophotometrically with an enzyme-linked immunosorbent assay (ELISA) kit (Beijing Winter Song Boye Biotechnology Co., Ltd., Beijing, China) following the kit’s protocol. Circulating immunoglobulin (IgA) (E101–102), IgG (E101–104) and IgM (E101–100) were quantified using porcine specific ELISA kits following the instructions of the kits (Bethyl Laboratories, Montgomery, TX, USA).

### 2.7. Analysis of Cytokines Concentration

The frozen jejunal and ileal mucosa samples were weighed (about 1.0 g), ground with a hand-held tissue homogenizer (Fisher, Pittsburgh, PA, USA) in 9 mL cold physiological saline and centrifuged at 4000 r/min for 10 min at 4 °C. The supernatants were harvested for detecting the level of cytokines using a microplate reader (BioRad, Hercules, CA, USA) with corresponding ELISA kits (Beijing 4A Biotech Co., Ltd., Beijing, China) following the kit instructions. The concentration of total protein in mucosa homogenates was tested with the BCA kit to normalize the cytokine level.

### 2.8. RT-PCR

Total RNA extraction, RT reactions and RT-PCR were conducted following the corresponding kit instructions. The primers utilized in this study were provided by Invitrogen (Shanghai, China) and presented in Table 2 *β-actin* served as the reference. Gene expression levels were analyzed following the 2^–ΔΔCT^ method [29].

### 2.9. Metachromatic ATPase Staining

ATPase staining was conducted as described previously [30]. In brief, the collected LT muscles were frozen in cooled isopentane near its freezing point and coated in OTC to obtain serial transverse sections (10 μm thick) with a cryostat (CM1850, Leica, Germany) at −20 °C. After a series of incubation, the fast-fibers were identified at pH 4.35.

### 2.10. Data Analysis

All data were analyzed by a one-way ANOVA procedure of SPSS 21.0 (SPSS, Inc., Chicago, IL, USA). Differences among treatments were tested using a Tukey multiple comparison test procedure of SPSS 21.0 (SPSS, Inc., Chicago, IL, USA). All results were presented as mean ± SD and *p* < 0.05 was regarded as statistically significant.

## 3. Results

### 3.1. Growth Performance

The effects of Eq supplementation on the growth performance of weanling piglets are shown in Table 3. During the experimental period, piglets in the IUGR group showed a decreased (*p* < 0.05) initial BW, final BW, ADG and ADFI when compared with piglets in the NBW group, while piglets in the IUGR + Eq group had a higher (*p* < 0.05) final BW, ADG and ADFI than those in the IUGR group.

### 3.2. Intestinal Morphology

As shown in Table 4, IUGR raised (*p* < 0.05) the duodenal crypt depth and decreased (*p* < 0.05) the duodenal and jejunal V/C and villus width, and the jejunal villus height as contrasted with the NBW group. Eq supplementation effectively improved (*p* < 0.05) the duodenal and jejunal V/C and villus width and jejunal villus height, and reduced (*p* < 0.05) the duodenal crypt depth of IUGR piglets.

### 3.3. Antioxidant Capacity

The redox statuses of serum and small intestines are exhibited in Table 5. In serum, IUGR piglets showed lower (*p* < 0.05) SOD and CAT activities and higher (*p* < 0.05) MDA content than those in NBW group. Eq supplementation improved (*p* < 0.05) the activities of SOD and CAT, and reduced (*p* < 0.05) the concentration of MDA in piglets with IUGR. In duodenum, piglets in the IUGR group showed lower (*p* < 0.05) GSH-Px, SOD and CAT activities and higher (*p* < 0.05) MDA levels than those in the NBW group. Eq supplementation remarkably enhanced (*p* < 0.05) the SOD and CAT activities of IUGR piglets. In jejunum, a decrease (*p* < 0.05) in GSH-PX and SOD activities and an increase (*p* < 0.05) in MDA levels were found in IUGR piglets as contrasted with those in the NBW group. After Eq supplementation, IUGR piglets showed increased (*p* < 0.05) SOD activity and reduced (*p* < 0.05) MDA levels. In ileum, IUGR improved (*p* < 0.05) MDA concentration as contrasted with the NBW group, while Eq treatment significantly reduced (*p* < 0.05) the MDA concentration in IUGR piglets.

### 3.4. Serum Parameters

The effects of Eq supplementation on the serum index of weanling piglets are shown in Table 6. Contrasted with the NBW group, an increase (*p* < 0.05) in the serum DAO activity, and in the concentrations of D-lactate and endotoxin, and a decrease (*p* < 0.05) in the serum IgG concentration were observed in the IUGR group. After Eq intervention, piglets in the IUGR group showed lower (*p* < 0.05) serum DAO activity and concentrations of serum D-lactate and endotoxin, as well as higher (*p* < 0.05) serum IgG content.

### 3.5. Intestinal Cytokines Concentrations

The concentrations of cytokines, including IFN-γ, TNF-α, IL-4, and IL-10, in jejunum and ileum are presented in Table 7. Piglets in the IUGR group exhibited higher (*p* < 0.05) concentrations of TNF-α and IL-10 in jejunum and ileum than those in the NBW group. After Eq administration, the jejunal TNF-α and IL-10 concentrations and ileal TNF-α concentrations were lower (*p* < 0.05) in the IUGR + Eq group than those in the IUGR group, while no differences (*p* > 0.05) were observed in the jejunal and ileal IFN-γ and IL-4 concentrations among treatments.

### 3.6. Intestinal Barrier-Related Genes Expression

The expression levels of gut barrier-related markers, including *ZO1*, *ZO2*, *CLDN1*, *CLDN2*, *OCLN*, *MUC1*, *MUC2* and *TFF3*, are shown in Figure 1. In jejunum, piglets subjected to IUGR showed a decline (*p* < 0.05) in the mRNA expressions of *ZO1*, *CLDN1*, *OCLN*, *MUC1* and *TFF3*, as contrasted with piglets in the NBW group, whereas Eq administration markedly elevated (*p* < 0.05) the mRNA levels of *CLDN1*, *OCLN* and *TFF3* in IUGR piglets. In ileum, a decrease (*p* < 0.05) in the mRNA levels of *ZO1*, *CLDN1*, *OCLN* and *MUC2* was found in IUGR piglets when compared with those in NBW piglets. However, Eq supplementation remarkably elevated (*p* < 0.05) the mRNA levels of the aforementioned genes in piglets with IUGR.

### 3.7. Skeletal Muscle Fiber Characteristics

Figure 2 presents the influences of Eq supplementation on muscle fiber type in weanling piglets. ATPase staining showed an obvious decrease (*p* < 0.05; Figure 2A,B) in the percentage of fast-fibers in the LT muscle of IUGR piglets, and RT-PCR results indicated that IUGR decreased (*p* < 0.05; Figure 2C) the mRNA expressions of *MyHC IIx* and *MyHC IIb* and elevated (*p* < 0.05; Figure 2C) the mRNA expression of *MyHC I* when compared with NBW piglets. As expected, the aforesaid indicators were reversed following Eq supplementation in IUGR piglets (*p* < 0.05; Figure 2B,C).

## 4. Discussion

Pigs with IUGR are normally characterized by low birth weight and poor postnatal growth [3,31]. It has been suggested that the nonreversible oxidative injury, feeblish intestinal function, reduced muscular growth potential and impaired endocrine status and nutrient metabolism leads to the growth check of IUGR piglets [1,26,32,33,34]. Consistently, our results showed that piglets with IUGR (birth weight: 0.89 ± 0.06 kg) exhibited a lower ADG and ADFI when contrasted with the NBW piglets (birth weight: 1.53 ± 0.09 kg). Supplementation with Eq increased the final BW, ADG and ADFI of IUGR by 8.39, 14.6 and 15%, which might result in a heavier body weight (about 8.0 kg) at market than IUGR piglets fed a basal diet [35]. As the precursor of Eq, daidzein was reported to have a beneficial effect on the growth performance of weanling piglets under normal or challenge conditions [7,36]; however, whether Eq can improve the growth performance of IUGR piglets has not been reported yet. As expected, supplementation with Eq remarkably promoted the final BW, ADG and ADFI of IUGR piglets when compared to IUGR piglets without Eq treatment. The above results indicated that Eq has a positive influence on relieving the growth retardation of weanling piglets induced by IUGR.

The intestinal tract is a vital place for the digestion, absorption, and metabolism of nutrients [37]. Intestinal morphology, including villus height, villus width, crypt depth and V/C, is commonly regarded as an important marker to measure intestinal development and function [38], while substantial evidence has verified that an impaired intestinal structure, such as crypt hyperplasia, villus shedding and mucosal atrophy, often occurs in IUGR weanling piglets and is characterized by worse absorptive and digestive functions, thus leading to growth retardation [32]. Here, Eq supplementation increased duodenal and jejunal V/C and villus width, and duodenal villus height, and decreased duodenal crypt depth in piglets with IUGR, which suggested that Eq could restore the impaired intestinal morphology caused by IUGR. Consistently, previous research in vitro showed that Eq pretreatment protects primary chicken intestinal epithelial cells from H_2_O_2_-induced death or debility [21]. Therefore, the improved intestinal morphology in the present study might contribute to the improvement of growth rate.

Oxidative stress refers to a state of imbalance between the production of and the capacity to scavenge ROS [39]. However, many studies have indicated that IUGR piglets are susceptible to oxidative injury by elevating MDA levels and impairing the antioxidant defense system [1,40]. In this situation, piglets are prone to suffer from villus atrophy, intestinal barrier disorder, infection and inflammation [41]. The dietary addition of antioxidant substances with capacities to clear free radicals and improve antioxidant ability may be beneficial to maintain the balance of redox status. A previous study demonstrated that Eq can offer strong protective effects on chicken IECs against H_2_O_2_-induced oxidative injury via promoting T-SOD activity and Nrf2 mRNA levels [21], and possesses the strongest antioxidant activity among the isoflavone-derived compounds [42]. In this study, we evaluated the redox status in the serum and small intestine through testing several antioxidant and oxidative indicators. Results showed that the activities of SOD and CAT, two crucial endogenous antioxidant enzymes involved in repairing oxidative injury [43], elevated in the serum, duodenum or jejunum of piglets in the IUGR + Eq group, suggested that the antioxidant ability of IUGR piglets can be elevated by Eq supplementation. Similarly, Choi [17] reported that the oral administration of Eq remarkably suppressed the biomarkers of oxidative stress (thiobarbituric acid-reactive substances value, carbonyl content, and serum 8-OH-dG) and increased T-SOD and CAT activities and their mRNA expression in the livers of mice. Meanwhile, we observed a decreased level of MDA, a vital indicator for lipid peroxidation, in the serum, jejunum and ileum of IUGR piglets following Eq supplementation. Studies in J774 macrophage cells revealed that Eq could block low-density lipoprotein (LDL) oxidation via suppressing superoxide radical production and promoting the free nitric oxide level [44], which suggested that Eq could relieve oxidative injury by the inhibition of lipid peroxidation. Moreover, lipid peroxidation might be involved in the turnover of enterocytes across the crypt–villus axis in the intestine [45], which may induce a worse intestinal morphology in IUGR group. Therefore, the promotion of antioxidant ability by Eq treatment might conduce a superior intestinal morphology. The mechanism for the antioxidant property of Eq probably relies on its special structure. Briefly, Eq serves as a hydrogen/electron donor to clear free radicals, thus inhibiting the oxidant reaction [46,47].

In addition to the absorptive and digestive functions, the intestinal tract also severs as an important physiological barrier between the body and the outer environment [48]. A previous study confirmed that IUGR could lead to intestinal barrier dysfunction in weaned piglets with increased bacterial translocation, poor intestinal barrier integrity, and disordered expression profiles of barrier-related protein [49]. In this study, IUGR piglets exhibited higher circulating DAO activity and D-lactate levels than NBW piglets. DAO, an intracellular enzyme, is produced in intestinal epithelia and mainly exists in cytoplasm, and D-lactate is a byproduct of gut bacteria [50]. Once the intestinal barrier function is impaired, DAO and D-lactate will be released into blood [51]. Thus, the D-lactate level and DAO activity in the serum normally serve as the key markers for assessing intestinal permeability [52]. In addition, the increased intestinal permeability would allow the bacteria-derived endotoxin and toxic macromolecules entry into the body [48]. In this study, a decrease in the serum endotoxin concentration in IUGR piglets was observed when contrasted with that in NBW piglets, which was in line with a previous study [53]. Our results indicated that IUGR exerts a negative influence on the intestinal integrity of piglets. As a strong antioxidant, Eq shows cell cycle regulatory activity [18] and a potent repair effect on intestinal morphological injury induced by oxidative stress [21]. Similarly, supplementation with Eq protected piglets from IUGR-induced damaged intestinal mucosal barrier function, as indicated by reduced serum DAO activity and the levels of D-lactate and endotoxin. The intestinal epithelial barrier is mainly constitutive of epithelial cells anchored by tight junctions, such as OCLN, CLDN families and ZOs, constituting a selective permeable barrier between epithelial cells [54]. The expression of tight junction protein is critical for sustaining intestinal barrier integrity. However, IUGR could induce continuous damage to the intestinal barrier function via reducing the mRNA levels of tight junction proteins in piglets [1,55]. In this study, the lower mRNA levels of *ZO1*, *CLDN1* and *OCLN* in the jejunum or ileum of IUGR piglets were observed. Thus, the higher intestinal permeability in IUGR piglets might be attributed to the lower mRNA expression of *ZO1*, *CLDN1* and *OCLN*. In line with the improved intestinal barrier integrity, supplementation with Eq could repair the impaired intestinal barrier function of piglets induced by IUGR. Consistently, an in vitro study showed that pre-treatment with Eq remarkably elevated the transcript abundance of *CLDN1* in H_2_O_2_ treated primary chicken intestinal epithelial cells [21]. In addition, intestinal epithelial cells were covered by a layer of mucins and trefoil peptides secreted by goblet cells to form a protective barrier [56]. Membrane-bound mucins (MUC1) and secreted mucins (MUC2) were recently shown to participate in mucosal repair [57,58]. The trefoil factor family 3 (TFF3), a small peptide, exhibits numerous biological properties in the regulation of inflammation, healing and differentiation [59,60] and is required for epithelial recovery [61], thus sustaining intestinal mucosal integrity. An in vivo study in rats demonstrated that IUGR could induce the impairment of mucosal maturation by decreasing the expression of *MUC2* and *TFF3* as contrasted with control pups [56]. In this study, the upregulation of *MUC1*, *MUC2* and *TFF3* in the jejunum or ileum of IUGR piglets following Eq supplementation may benefit the intestinal barrier function and restore mucosal injuries. Therefore, the results from this study, both direct and indirect evidence, suggest that Eq could promote the intestinal barrier function of IUGR piglets partly via upregulating intestinal-barrier-related genes expression.

The levels of serum immunoglobins are considered to be vital markers to evaluate immune status [55]. In this study, the serum IgG level was reduced in IUGR piglets, which was consistent with the study of Che et al. [62]. Interestingly, Eq supplementation significantly elevated the serum IgG concentration in IUGR piglets. In addition, increasing evidence has verified that inflammation is a primary inducible factor in intestinal barrier disorder [54,63]. Cytokines perform vital roles in the immune and inflammatory responses [64]. Proinflammatory cytokines, TNF-α and IFN-γ, participate in the inflammatory process and are critical for the initiation of the inflammatory response when animals are infected, whereas the overproduction of proinflammatory cytokines leads to a serious inflammatory response and directly disrupts the epithelial barrier integrity [65,66]. IUGR can cause intestinal inflammatory injury by increasing proinflammatory cytokines content and their mRNA expressions [1]. In this study, the injured intestinal barrier function in IUGR piglets might be partly attributed to the higher TNF-α level in the jejunum and ileum. On the other hand, anti-inflammatory cytokines IL-4 and IL-10 block the overactivation of the immune response and inhibit the overproduction of proinflammatory cytokines to control immune homeostasis [67]. Thus, we speculated that the higher IL-10 levels in the jejunum and ileum of IUGR piglets was vital for keeping the balance of the immune system. In the present study, decreases in IL-4 and IL-10 levels in the jejunum or ileum of IUGR piglets were found in response to Eq treatment, which was in line with the results in chicken HD11 macrophages [16]. The mechanisms for the lower TNF-α and IL-0 may rely on the protective effect of Eq in preventing the immune system from overactivation in IUGR piglets. In addition, avoiding the overstimulation of the immune system could save unnecessary nutrient consumption, which may partly contribute to greater growth performance.

Skeletal muscle commonly accounts for 35–40% of the body weight in neonates and the improvement of skeletal muscle development is of great significance for later growth and metabolism [2]. IUGR hurts the postnatal compensatory growth and development of skeletal muscle and decreases the amount of fast-fiber and total muscle fiber [68], which potentially leads to reduced muscle mass and delayed muscle maturity [69]. Consistently, we observed that IUGR significantly reduced the percentage of fast-fibers and the mRNA expressions of *MyHC IIx* and *MyHC IIb*, and increased the mRNA expression of *MyHC I* in the LT muscle of IUGR piglets. A previous study demonstrated that the in ovo injection of Eq remarkably improved muscle water-holding ability, which was related to the improved antioxidant capacity in broilers [19]. Here, Eq supplementation increased the percentage of fast-fibers and the mRNA expression of *MyHC IIb*, which may contribute to an improved growth rate.

## 5. Conclusions

Collectively, the present study offers novel insights into the beneficial influence of Eq treatment on the growth performance of IUGR piglets, and the mechanisms might be attributed to better intestinal morphology, increased antioxidant capacity, improved intestinal barrier function, reduced intestinal inflammatory damage and promoted skeletal muscle development.

## Figures and Tables

**Figure 1 animals-13-01469-f001:**
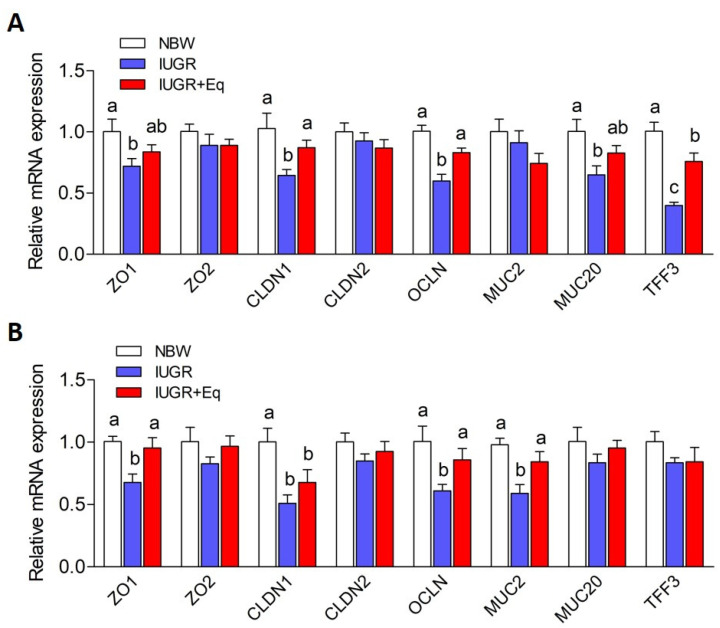
Effects of Eq supplementation on the mRNA level of ZO1, ZO2, CLDN1, CLDN2, OCLN, MUC2, MUC20 and TFF3 in jejunum (**A**) and ileum (**B**) of weanling piglets with IUGR. *β-actin* was considered as an internal reference gene. Data were exhibited as the means ± SD, n = 10/group. NBW group: NBW piglets were offered a basal diet; IUGR group: IUGR piglets were offered a basal diet; IUGR + Eq: IUGR piglets were offered a basal diet supplemented with 50 mg of Eq per kg of diet. *ZO1*, zona occludens 1; *ZO2*, zona occludens 2; *CLDN1*, claudin 1; *CLDN2*, claudin 2; OCLN, occluding; *MUC1*, mucin 1; *MUC2*, mucin 2; *TFF3*, trefoil factor family 3. The different letters indicate significant differences among treatments (*p* < 0.05).

**Figure 2 animals-13-01469-f002:**
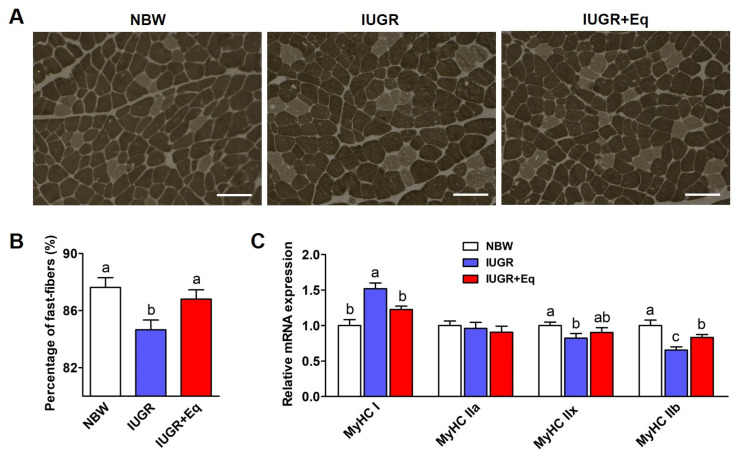
Effects of Eq supplementation on skeletal muscle fiber type in the LT muscle of weanling piglets with IUGR. (**A**) The ATPase staining of LT muscles. Fast-fibers stain dark gray. Original magnification, ×100. Bars: 100 μm. (**B**) The percentage of fast-twitch fibers was analyzed based on the ATPase staining. (**C**) Relative mRNA expressions of four *MyHC* isoforms in LT muscles. *β-actin* was used as an internal reference gene. Data were exhibited as the means ± SD, n = 10/group. NBW group: NBW piglets were offered a basal diet; IUGR group: IUGR piglets were offered a basal diet; IUGR + Eq: IUGR piglets were offered a basal diet supplemented with 50 mg of Eq per kg of diet. *MyHC*, myosin heavy chain. The different letters indicate significant differences among treatments (*p* < 0.05).

**Table 1 animals-13-01469-t001:** Composition and nutrient level of basal diet (as-fed basis).

Ingredients	Content (%)
Corn (crude protein 7.8%)	28.50
Extruded corn (crude protein 8.2%)	25.45
Soybean meal (crude protein 44.2%)	11.50
Extruded soybean	10.50
Fish meal (crude protein 62.5%)	4.00
Whey powder (low protein)	7.00
Soybean protein concentrate	5.00
Soybean oil	1.50
Glucose	4.00
Limestone	0.65
Dicalcium phosphate	0.60
Salt	0.30
L-Lysine HCl (78%)	0.30
DL-Methionine	0.08
L-Threonine (98.5%)	0.06
Tryptophan	0.02
Chloride choline	0.10
Vitamin premix [26]	0.04
Mineral premix [26]	0.40
Total	100.00
Calculated composition	
Crude protein (%)	19.20
Digestive energy (MJ/kg)	14.86
Calcium (%)	0.92
Total phosphorus (%)	0.55
Available phosphorus (%)	0.37
SID Lysine (%)	1.32
SID Methionine (%)	0.44
SID Threonine (%)	0.76
SID Methionine + cysteine (%)	0.72
SID Tryptophan (%)	0.20
Crude fiber (%)	2.46
Soluble fiber (%)	1.31

**Table 2 animals-13-01469-t002:** Primers for real-time quantitative PCR.

Gene	Primer Sequence (5′→3′)	Accession No.
*MyHC-I*	Forward: GTTTGCCAACTATGCTGGGG	XM_021098827.1
	Reverse: TGTGCAGAGCTGACACAGTC	
*MyHC-IIa*	Forward: CTCTGAGTTCAGCAGCCATGA	NM_001244539.1
	Reverse: GATGTCTTGGCATCAAAGGGC	
*MyHC-IIx*	Forward: TTGACTGGGCTGCCATCAAT	NM_011636347.2
	Reverse: GCCTCAATGCGCTCCTTTTC	
*MyHC-IIb*	Forward: GAGGTACATCTAGTGCCCTGC	XM_003357928.4
	Reverse: GCAGCCTCCCCAAAAATAGC	
*ZO1*	Forward: GCCAGCTGGAGCTTAGAACA	XM_021098896.1
	Reverse: GGCATCAAGAGGGGCTACTG	
*ZO2*	Forward: TCTTCCTTTGAGGCTCTTTACA	NM_001206404.1
	Reverse: TCTTCGGGCTGTTTAGCACT	
*CLDN1*	Forward: ACAGGAGGGAAGCCATTTTCA	NM_001244539.1
	Reverse: TTTAAGGACCGCCCTCTCCC	
*CLDN2*	Forward: ATCTAGCGCCATCTCCTCGT	XM_021079578.1
	Reverse: TCTTTGGCGCGAGAGTTCTG	
*OCLN*	Forward: CAGGTGCACCCTCCAGATTG	NM_001163647.2
	Reverse: ATGTCGTTGCTGGGTGCATA	
*MUC1*	Forward: AGATCCCACCACCAGCTACT	XM_021089729.1
	Reverse: AAAAGAGTCCCAGAAGCCCG	
*MUC2*	Forward: GGGGCGACTTCCACTACAAG	XM_021082584.1
	Reverse: CCGTCAGCAGGATGTACTCG	
*TFF3*	Forward: GAGTCATGCTCGCTCCCTTT	NM_001243483.1
	Reverse: GCTTCTCAAGGGTCACGGAA	
*β-actin*	Forward: TCCATCGTCCACCGCAAATG	XM_003357928.4
	Reverse: TTCAGGAGGCTGGCATGAGG	

*MyHC*, myosin heavy chain; *ZO1*, zona occludens 1; *ZO2*, zona occludens 2; *CLDN1*, claudin 1; *CLDN2*, claudin 2; *OCLN*, occluding; *MUC1*, mucin 1; *MUC2*, mucin 2; *TFF3*, trefoil factor family 3.

**Table 3 animals-13-01469-t003:** Effects of Eq supplementation on the BW, ADG, ADFI, and F/G of weanling piglets with IUGR (means ± SD, n = 10/group).

Item ^2^	Treatment ^1^			*p*-Value
	NBW	IUGR	IUGR + Eq	
Initial BW, kg	6.02 ± 0.20 ^a^	4.45 ± 0.18 ^b^	4.50 ± 0.15 ^b^	0.029
Final BW, kg	12.82 ± 0.32 ^a^	9.53 ± 0.22 ^c^	10.33 ± 0.24 ^b^	0.043
ADG, g/d	242.44 ± 7.23 ^a^	181.56 ± 4.55 ^c^	208.07 ± 5.30 ^b^	0.008
ADFI, g/d	398.27 ± 14.09 ^a^	300.41 ± 11.06 ^c^	345.48 ± 15.67 ^b^	0.036
F/G	1.64 ± 0.02	1.65 ± 0.03	1.67 ± 0.03	0.831

^1^ NBW group: NBW piglets were offered a basal diet; IUGR group: IUGR piglets were offered a basal diet; IUGR + Eq: IUGR piglets were offered a basal diet supplemented with 50 mg of Eq per kg of diet. ^2^ BW, body weight; ADG, average daily gain; ADFI, average daily feed intake; F/G, feed to gain ratio. The different letters indicate significant differences among treatments (*p* < 0.05).

**Table 4 animals-13-01469-t004:** Effects of Eq supplementation on villus height, crypt depth, V/C, villus width, and villus surface area of weanling piglets with IUGR (means ± SD, n = 10/group).

Item ^2^	Treatment ^1^			*p*-Value
	NBW	IUGR	IUGR + Eq	
Duodenum				
Villus height, μm	427.38 ± 41.13	365.81 ± 35.56	389.24 ± 30.20	0.106
Crypt depth, μm	219.66 ± 20.07 ^b^	263.32 ± 26.40 ^a^	208.79 ± 19.81 ^b^	0.018
V/C	1.94 ± 0.17 ^a^	1.39 ± 0.13 ^b^	1.86 ± 0.21 ^a^	0.006
Villus width, μm	138.29 ± 5.22 ^a^	103.46 ± 3.82 ^c^	119.66 ± 4.35 ^b^	0.044
VSA, mm^2^	0.094 ± 0.05	0.060 ± 0.01	0.073 ± 0.02	0.669
Jejunum				
Villus height, μm	364.50 ± 31.41 ^a^	322.58 ± 24.47 ^b^	386.30 ± 29.19 ^a^	0.037
Crypt depth, μm	172.18 ± 12.80	198.09 ± 15.88	189.13 ± 20.25	0.208
V/C	2.18 ± 0.27 ^a^	1.64 ± 0.19 ^b^	2.04 ± 0.25 ^a^	0.005
Villus width, μm	121.49 ± 4.66 ^a^	98.38 ± 2.15 ^c^	108.19 ± 3.36 ^b^	0.009
VSA, mm^2^	0.098 ± 0.04	0.050 ± 0.01	0.066 ± 0.01	0.415
Ileum				
Villus height, μm	240.23 ± 20.95	223.51 ± 16.28	246.17 ± 30.67	0.553
Crypt depth, μm	168.83 ± 19.60	166.04 ± 21.37	177.34 ± 17.44	0.751
V/C	1.42 ± 0.30	1.35 ± 0.16	1.39 ± 0.23	0.419
Villus width, μm	110.05 ± 3.11	104.34 ± 4.03	106.14 ± 2.89	0.606
VSA, mm^2^	0.043 ± 0.01	0.037 ± 0.01	0.041 ± 0.01	0.844

^1^ NBW group: NBW piglets were offered a basal diet; IUGR group: IUGR piglets were offered a basal diet; IUGR + Eq: IUGR piglets were offered a basal diet supplemented with 50 mg of Eq per kg of diet. ^2^ V/C, villus height to crypt depth ratio; VSA, villus surface area. The different letters indicate significant differences among treatments (*p* < 0.05).

**Table 5 animals-13-01469-t005:** Effects of Eq supplementation on the activities of GSH-Px, T-AOC, SOD and CAT, and the content of MDA in serum and intestines of weanling piglets with IUGR (means ± SD, n = 10/group).

Item ^2^	Treatment ^1^			*p*-Value
	NBW	IUGR	IUGR + Eq	
Serum				
GSH-Px, U/mL	0.30 ± 0.07	0.23 ± 0.06	0.27 ± 0.07	0.772
T-AOC, U/mL	2.12 ± 0.15	1.86 ± 0.23	1.97 ± 0.17	0.609
SOD, U/mL	12.67 ± 2.17 ^a^	7.45 ± 0.98 ^b^	10.76 ± 1.59 ^a^	0.039
MDA, nmol/mL	2.11 ± 0.31 ^b^	3.10 ± 0.44 ^a^	2.45 ± 0.38 ^b^	0.043
CAT, U/mL	15.27 ± 1.81 ^a^	12.82 ± 1.57 ^b^	16.45 ± 2.09 ^a^	0.029
Duodenum				
GSH-Px, U/mg prot	4.35 ± 0.45 ^a^	3.25 ± 0.37 ^b^	3.81 ± 0.62 ^b^	0.048
T-AOC, U/mg prot	0.48 ± 0.10	0.40 ± 0.07	0.45 ± 0.08	0.731
SOD, U/mg prot	16.33 ± 2.02 ^a^	10.80 ± 0.96 ^b^	15.09 ± 1.73 ^a^	0.024
MDA, nmol/mg prot	3.26 ± 0.32 ^b^	4.05 ± 0.42 ^a^	3.64 ± 0.36 ^ab^	0.046
CAT, U/g prot	7.51 ± 0.61 ^a^	5.52 ± 0.57 ^b^	8.58 ± 0.93 ^a^	0.038
Jejunum				
GSH-Px, U/mg prot	3.65 ± 0.31 ^a^	2.61 ± 0.27 ^b^	3.18 ± 0.34 ^ab^	0.044
T-AOC, U/mg prot	0.60 ± 0.18	0.49 ± 0.20	0.57 ± 0.16	0.786
SOD, U/mg prot	19.45 ± 2.16 ^a^	13.87 ± 1.78 ^b^	20.45 ± 1.91 ^a^	0.036
MDA, nmol/mg prot	2.38 ± 0.28 ^b^	3.58 ± 0.31 ^a^	2.72 ± 0.30 ^b^	0.041
CAT, U/g prot	6.04 ± 0.43	5.37 ± 0.48	5.68 ± 0.37	0.881
Ileum				
GSH-Px, U/mg prot	2.58 ± 0.25	2.21 ± 0.31	2.46 ± 0.22	0.692
T-AOC, U/mg prot	0.59 ± 0.12	0.52 ± 0.09	0.56 ± 0.10	0.773
SOD, U/mg prot	21.14 ± 2.72	17.58 ± 2.03	19.11 ± 1.96	0.301
MDA, nmol/mg prot	2.43 ± 0.21 ^c^	4.37 ± 0.30 ^a^	3.46 ± 0.25 ^b^	0.032
CAT, U/g prot	5.04 ± 0.51	4.44 ± 0.40	3.91 ± 0.43	0.205

^1^ NBW group: NBW piglets were offered a basal diet; IUGR group: IUGR piglets were offered a basal diet; IUGR + Eq: IUGR piglets were offered a basal diet supplemented with 50 mg of Eq per kg of diet. ^2^ GSH-Px, glutathione peroxidase; T-AOC, total antioxidant capacity; SOD, superoxide dismutase; MDA, malondialdehyde; CAT, catalase. The different letters indicate significant differences among treatments (*p* < 0.05).

**Table 6 animals-13-01469-t006:** Effects of Eq supplementation on serum DAO, D-lactate, endotoxin and immunoglobins of weanling piglets with IUGR (means ± SD, n = 10/group).

Item ^2^	Treatment ^1^			*p*-Value
	NBW	IUGR	IUGR + Eq	
DAO, U/L	7.52 ± 0.62 ^b^	10.23 ± 0.50 ^a^	8.26 ± 0.45 ^b^	0.025
D-lactate, μg/mL	23.06 ± 1.44 ^c^	27.11 ± 1.05 ^a^	25.28 ± 0.89 ^b^	0.041
Endotoxin, EU/mL	153.04 ± 9.57 ^b^	184.17 ± 12.49 ^a^	160.30 ± 10.77 ^b^	0.029
IgA, g/L	0.31 ± 0.03	0.33 ± 0.03	0.29 ± 0.02	0.882
IgM, g/L	0.79 ± 0.05	0.77 ± 0.07	0.82 ± 0.06	0.789
IgG, g/L	6.24 ± 0.38 ^a^	5.58 ± 0.42 ^b^	6.82 ± 0.56 ^a^	0.039

^1^ NBW group: NBW piglets were offered a basal diet; IUGR group: IUGR piglets were offered a basal diet; IUGR + Eq: IUGR piglets were offered a basal diet supplemented with 50 mg of Eq per kg of diet. ^2^ DAO, diamine oxidase; Ig, immunoglobulin. The different letters indicate significant differences among treatments (*p* < 0.05).

**Table 7 animals-13-01469-t007:** Effects of Eq supplementation on the concentrations of IFN-γ, TNF-α, IL-4 and IL-10 of weanling piglets with IUGR (means ± SD, n = 10/group).

Item ^2^	Treatment ^1^			*p*-Value
	NBW	IUGR	IUGR + Eq	
Jejunum				
IFN-γ, pg/mg of prot	97.16 ± 10.02	107.33 ± 9.47	102.14 ± 12.41	0.692
TNF-α, pg/mg of prot	55.23 ± 1.98 ^c^	67.33 ± 2.40 ^a^	59.12 ± 2.13 ^b^	0.029
IL-4, pg/mg of prot	81.76 ± 4.33	80.14 ± 5.01	77.58 ± 3.69	0.811
IL-10, pg/mg of prot	24.75 ± 2.09 ^b^	31.22 ± 2.27 ^a^	26.05 ± 1.87 ^b^	0.011
Ileum				
IFN-γ, pg/mg of prot	87.68 ± 7.25	95.04 ± 8.44	83.23 ± 10.28	0.558
TNF-α, pg/mg of prot	63.12 ± 2.65 ^b^	79.67 ± 4.44 ^a^	67.16 ± 3.35 ^b^	0.008
IL-4, pg/mg of prot	95.43 ± 5.61	102.11 ± 6.22	93.79 ± 6.88	0.723
IL-10, pg/mg of prot	33.07 ± 2.75	40.91 ± 3.44	37.18 ± 3.14	0.085

^1^ NBW group: NBW piglets were offered a basal diet; IUGR group: IUGR piglets were offered a basal diet; IUGR + Eq: IUGR piglets were offered a basal diet supplemented with 50 mg of Eq per kg of diet. ^2^ IFN-γ, interferon-γ; TNF-α, tumor necrosis factor-α; IL-4, interleukin-4; IL-10, interleukin-10. The different letters indicate significant differences among treatments (*p* < 0.05).

## Data Availability

The data in this study are available on reasonable request.

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
