# Peer review of "Effects of Equol Supplementation on Growth Performance, Redox Status, Intestinal Health and Skeletal Muscle Development of Weanling Piglets with Intrauterine Growth Retardation"

_animals, 2023, doi:10.3390/ani13091469_

Round 1
Reviewer 1 Report
I have reviewed the article entitled: Effects of equol supplementation on growth performance, redox status, intestinal health and skeletal muscle development of weanling piglets with intrauterine growth retardation
The research is sound but several issues must be clarified
In the body of the manuscript, the name of all abreviations should be spelled the first time the appear., even if they first appear in the Abstract. See IUGR in the Introduction.
The dietary amino acid content should be given in digestible basis
In the design, explain why the NBW+Eq treatment was not included since all piglets at weaning face different stress factors not only IUGR piglets, hence, Eq can also improve the performance of all kind of pigs at weaning.
Explain why, the redox status, intestinal health and skeletal muscle development were not taken at weaning to find differences due to IUGR before piglets were subjected to the stress of weaning.
In the Results, for each variable response, the comparisons between the NBW vs IUGR should be given first, and then the comparison between IUGR vs IUGR+Eq; in most cases the results of NBW piglets are omitted.
In Tables 3, 4 and 5 use appropriate superscripts. The use of *, ** is confusing, and the NBW is lacking of superscripts. All treatments should have a superscript to improve the understanding of the results.
Explain why the area of the villi was not evaluated since it expresses better the digestion and absorption capacity of the mucosa. Villi can be taller but thinner or they can be shorter but thicker. Unfortunately the thickness of the villi was not measured.
Other corrections to be made:
In line 180, remove "with IUGR".
In line 226, correct "seerum"
In line 227 and 277, remove "IUGR"
In line 181 remove, "in"
English must be corrected since there are several written mistakes.
Reviewer 3 Report
The manuscript presented for review is very interesting and the presented analyzes are conducted in many directions.
Here are some considerations to consider.
Line 93 please explain what was the random assignment to the NBW group? as well as Line 94.
Line 107 - please describe exactly what the term means - 1, 28 I understand that there are days of experience and not the age of the piglets. I believe that you can put the age of the piglets in parentheses.
The methodological part lacks information on the preparation of tissue homogenates.
Please provide details of the apparatus used for the marking (symbol, company, country...).
I am asking for a more detailed description of statistical methods (statistical model).
There is no information on the health status of the animals involved in the experiment. Were there any diarrhea, treatment, falls. Please complete it.
Please present in Tables 3-7 the significance of differences as Pvalue and not as "*". I suggest to include Pvalue columns and remove "*"
Line 298, 299 - piglet weight in brackets misleads the reader - please verify - it may be described differently (e.g. birth weight).
After entering comments, the manuscript can be published.
Round 2
Reviewer 1 Report
You did a nice study.
Please add an appropriate reference for the VSA calculation formula.
